# Representations in human primary visual cortex drift over time

Zvi N. Roth [1] ✉ & Elisha P. Merriam [1]

Primary sensory regions are believed to instantiate stable neural representations, yet a number of recent rodent studies suggest instead that representations drift over time. To test whether sensory representations are stable in human visual cortex, we analyzed a large longitudinal dataset of fMRI responses to images of natural scenes. We fit the fMRI responses using an image-computable encoding model and tested how well the model generalized across sessions. We found systematic changes in model fits that exhibited cumulative drift over many months. Convergent analyses pinpoint changes in neural responsivity as the source of the drift, while population-level representational dissimilarities between visual stimuli were unchanged. These observations suggest that downstream cortical areas may read-out a stable representation, even as representations within V1 exhibit drift.

A guiding principle in sensory neuroscience is that neural representations of sensory information are stable, that a given stimulus will reliably evoke a particular pattern of neural activity. This assumption is so deeply ingrained in contemporary neuroscience that it has become implicit in many theories of neural coding. Similarly, a number of prominent theoretical frameworks assume that sensory signals are initially encoded via stable computations, which then allow downstream brain areas to decode those representations, perform additional transformations, reach a decision, and ultimately generate a motor output.

But recent electrophysiology studies have found that neural representations of natural stimuli in mouse visual cortex are not stable, and instead change, or drift, at timescales ranging from minutes to weeks[1–4]. The discovery of this phenomenon, termed 'representational drift', was made possible by technology that enabled large-scale chronic recordings of populations of neurons in rodent cortex over extended periods of time. An obvious barrier to addressing similar questions in primates is the dearth of comparable long-term recordings in primate visual cortex (but see[5–7]). Hence, it is not known whether representational drift occurs in human visual cortex, since up until now, no dataset has afforded the requisite longitudinal measurements.

Uncovering the mechanism of representational drift can guide in identifying principles underlying neural coding. Models of neural coding invariably assume that certain aspects of neural responses are critical for representing information while other aspects are redundant or irrelevant. For example, according to the labeled-line framework, the identity of the neurons most active within a given population is critical for reliably representing a stimulus, while the pattern of activity of other neurons in the population is not relevant for the neural representation. Conversely, if information is represented by an ensemble code, only the mean activity may be important, while the activity of individual neurons is of little or no consequence for downstream brain regions. Finally, according to models of population coding, neural representations are embedded in the pattern of activity across a population of neurons, while the absolute firing rate, both of individual neurons and the mean across the population, is not relevant. Each of these theories makes distinct predictions regarding the aspects of neural responses that must remain fixed and the aspects that may change (or drift) over time while still maintaining a sensory representation. Identifying the features of neural tuning that change versus those that remain stable, therefore, may ultimately reveal the most basic principles of neural coding.

## Results

We analyzed fMRI BOLD activity collected while subjects viewed a large database of naturalistic scenes in many scanning sessions over the course of a year[8]. To enable comparison of representations across sessions, we used an encoding model to estimate tuning properties within V1 during each individual session (Fig. 1A). We then tested whether tuning changed over time. If representations are stable, the model fit on data from one session should be equally good at predicting responses from any other session. But we found, on the

[1]Laboratory of Brain and Cognition, National Institute of Mental Health, NIH, Bethesda, MD, USA. ✉e-mail: zvi.roth@nih.gov

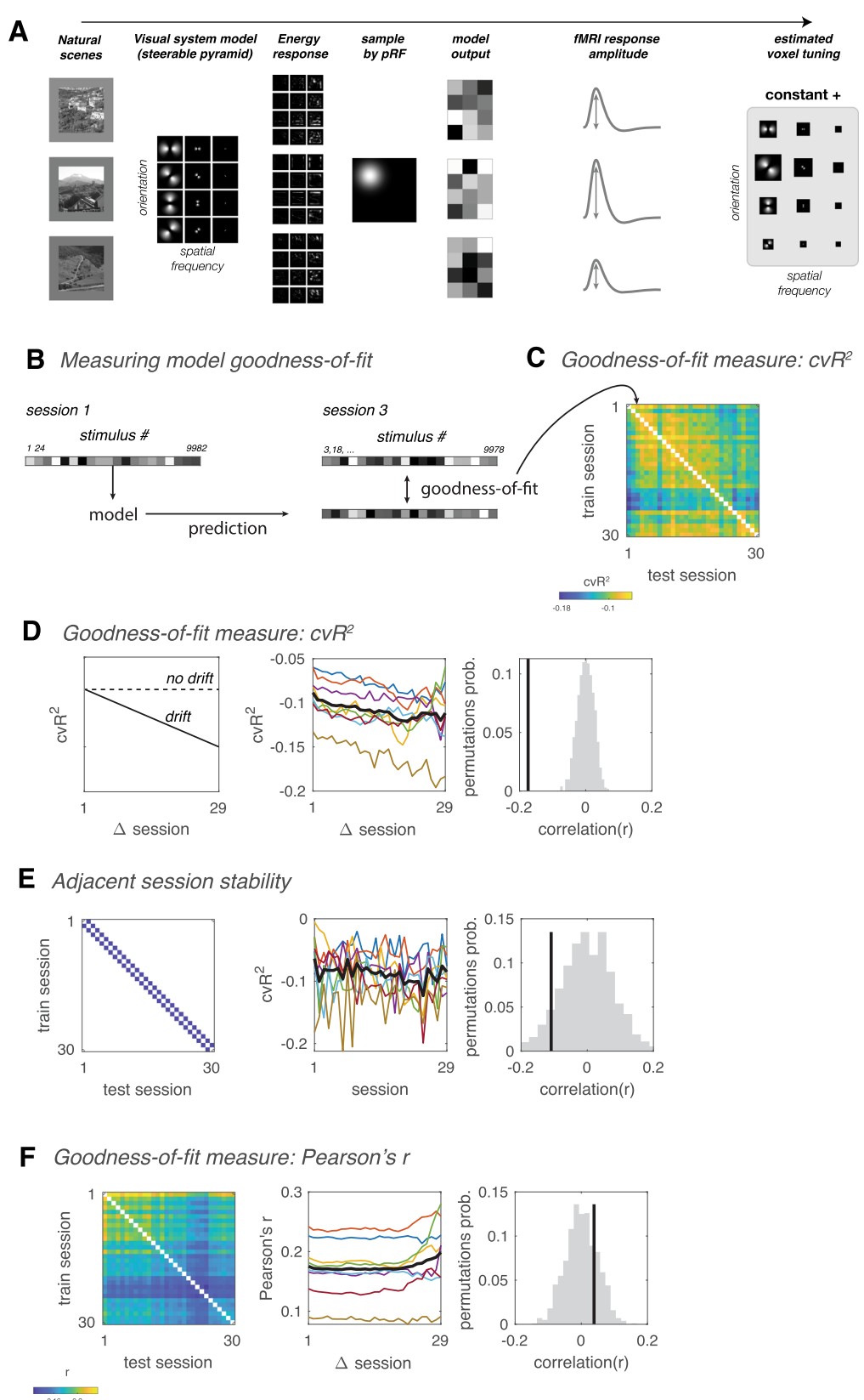

contrary, that the model's goodness-of-fit, measured by cross-validated $R^2$ ($cvR^2$), dropped with the number of sessions intervening between train and test sessions (Fig. 1C, D). Model generalization across adjacent pairs of sessions, on the other hand, did not change across the experiment, indicating that signal-to-noise ratio (SNR) was stable, and that the drift that we observed was not simply the result of a change in measurement reliability (Fig. 1E) or due to changes between the first and second sessions (Fig. S1). These analyses suggest that the representational drift was due to progressive changes in a neural process that continued throughout the year-long experiment.

Drift was evident in the data using $cvR^2$ as a measure of cross-session generalizability (Fig. 1C), but surprisingly, not when using

**Fig. 1 | Changes in cross-session generalization indicate representational drift.**
**A** Model-fitting pipeline for a single fMRI voxel timeseries measurement. Input consisted of images viewed by a particular subject in a particular session. Filter outputs were sampled by the pRF. The model assigns weights for each orientation and spatial frequency filter by multiple linear regression, using model outputs to predict response amplitudes. Example images shown here were created by the authors for illustration only and were not used in the study. **B** The model is trained independently on each session, and then tested on each other session. **C** Goodness-of-fit matrix, quantified by cross-validated $R^2$ (cv$R^2$), testing how well the model trained on each session predicted responses in all other sessions. Different diagonals of the matrix correspond to different numbers of intervening sessions between training and testing. Goodness-of-fit matrices were computed for each subject as the median of all V1 voxels, and averaged across subjects.
**D** Representations drift across sessions when quantified by cv$R^2$. Left, Schematic illustrating representational drift (solid line, cv$R^2$ decreases systematically with number of intervening sessions), and representational stability (dotted line, cv$R^2$ remains constant). Middle, Mean cv$R^2$ as function of number of intervening sessions between train and test sessions. Colored lines, individual subjects; black line,

mean across subjects. Predictive power of models decreases with number of intervening sessions, indicating representational drift ($r = -0.17$, $p < 0.001$). Drift was significant ($p < 0.05$) for all 8 individual subjects. Right, Black vertical line, empirical correlation between goodness-of-fit and number of intervening sessions. Gray histogram, null distribution of correlation values computed by randomizing the order of sessions 1000 times. Correlation was computed using all off-diagonal matrix values for each subject, and averaged across subjects. **E** Signal-to-noise ratio does not consistently decrease (or increase) across sessions. Left, Diagonals of the goodness-of-fit matrix corresponding to training and testing on adjacent sessions. Middle, Performance of model trained and tested on adjacent sessions, as function of earlier session of the two. Model performance does not systematically decrease across sessions ($r = -0.11$, $p = 0.086$). Right, Black vertical line, empirical correlation between adjacent-session performance and number of intervening sessions. Gray histogram, null distribution of correlation values. **F** Same as **B**, quantifying goodness-of-fit with Pearson's correlation instead of cv$R^2$ values. With this measure predictive power does not decrease with time ($r = 0.04$, $p = 0.774$). Source data are provided as a Source Data file.

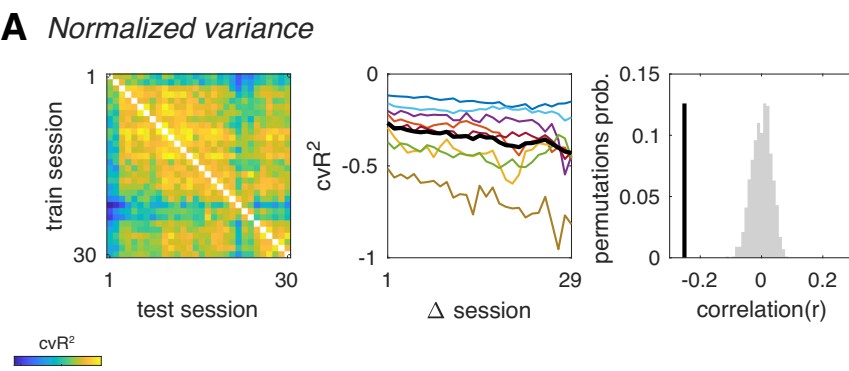

**A** *Normalized variance*

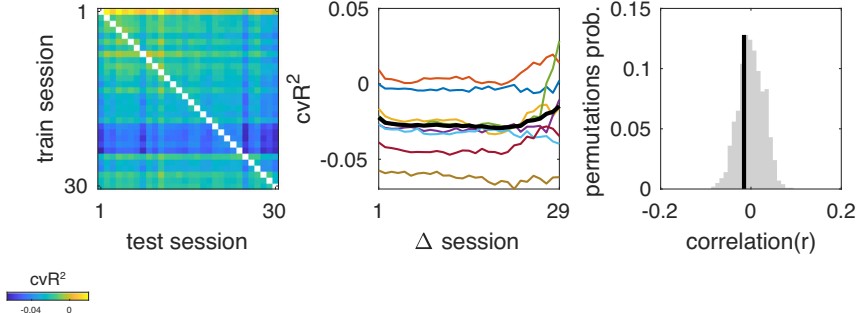

**B** *Normalized mean amplitude*

**Fig. 2 | Normalizing response amplitude, but not variance, removes drift.**
**A** Cross-session generalization after normalizing each session's variance. Left, goodness-of-fit matrix after normalizing each voxel's response variance within each session. Center, Mean cv$R^2$ as function of number of intervening sessions between train and test sessions. Model predictive power decreases with time, indicating representational drift ($r = -0.25$, $p < 0.001$). Gray lines, individual subjects; thick black line, mean across subjects. Right, Black vertical line, empirical correlation between goodness-of-fit and number of intervening sessions. Gray histogram, null

distribution of correlation values. **B** Cross-session generalization after subtracting each session's mean response amplitude (i.e. mean beta). Left, goodness-of-fit matrix after subtracting each voxel's mean response within each session. Center, Mean cv$R^2$ as function of number of intervening sessions between train and test sessions. After subtracting each voxel's mean, predictive power of V1 models no longer decrease with time ($r = -0.02$, $p = 0.287$). Source data are provided as a Source Data file.

correlation (Fig. 1F). The divergence between cv$R^2$ and correlation measures suggests that the drift is due to changes in either the mean and/or variance of response amplitudes, since cv$R^2$ is sensitive to both the mean and variance, whereas correlation is not. To test whether drift is caused by changes in the mean and/or variance, we equalized either the mean or the variance of each voxel's response amplitudes within each session and then recomputed the cross-session goodness-of-fit matrices. Whereas equalizing the variance had a minimal impact

on drift (Fig. 2A), removing the mean response amplitude significantly attenuated representational drift (Fig. 2B) We also found that response amplitudes exhibited dependencies across sessions, evident in positive autocorrelations across several sessions (Fig. S2). These observations indicate that response amplitudes of individual fMRI voxels gradually accumulated changes across sessions and were not simply fluctuating around a fixed mean, as would be expected from random noise. Our observations are consistent with primate studies at much

shorter timescales[9,10], and suggest that drift in human V1 is primarily due to inter-session changes in mean response amplitude.

What changes in the fMRI response tuning resulted in the gradual changes in mean response amplitude? Several properties of tuning functions could potentially change with time, resulting in a change in BOLD fMRI responses (Fig. S3), including changes in baseline, gain, tuning width, and stimulus preference. However, simulations suggest that only an additive baseline shift would affect the mean response amplitude without a concomitant change in the variance of responses, consistent with the representational drift that we observed. Consistent with this scenario, we found that the baseline coefficient exhibited positive autocorrelations (Fig. S4), indicating cumulative changes across sessions, and suggesting that the drift reflects systematic shifts in the baseline.

Drift of stimulus representations locally within individual neurons could potentially create a conundrum for the brain since it implies unreliable information coding. But activity in downstream brain areas may depend on a read-out mechanism that takes into account the entire population of responses across V1. Such a mechanism would be advantageous because it would be robust to changes in individual neurons. We therefore asked: does the population representation drift as well? One possibility is that all of V1 undergoes a uniform additive change, and as a result, representations across populations of responses in V1 maintain a fixed spatial profile over time, up to an additive shift. If this is the case, similarity between representations at different time points should remain roughly constant. However, cross-correlations between voxels indicate that, on the contrary, inter-session changes were not uniform across V1; While some voxels exhibited changes that were positively correlated with one another, other voxels were anticorrelated (Fig. 3E), suggesting that distinct subregions of cortex undergo different changes in mean response amplitude. If the changes are spatially heterogeneous, then as drift accumulates across sessions, population responses should become increasingly different from the original pattern. To test this, we estimated population responses to a range of stimuli, using the encoding model weights estimated from each session (see *Methods: Simulating population responses*). We examined how single images are represented by the entire V1 population by measuring the correlation between representations in different sessions (Fig. 3A). If all of V1 experiences the same baseline shift, then these shifts should not cause a decrease in inter-session correlations, since correlation is not sensitive to additive changes. We observed a decreasing similarity between population representations as the number of intervening sessions increased (Fig. 3B–D). This result implies that population-level representations gradually drifted across sessions.

If population representations of stimuli drift over time, how does the visual system maintain a stable perception of the world? We hypothesized that while responses to individual stimuli undergo changes, their representations remain stable relative to other stimuli. To test this possibility, we constructed a representational dissimilarity matrix (RDM) for each session, reflecting the dissimilarity between population responses to a range of images (Fig. 3F). We found that despite ongoing changes in single image representations, correlations between RDMs did not diminish over time (Fig. 3G–I), indicating that the RDM remained stable across sessions. This suggests that, while the representation itself is dynamic, the position of representations in stimulus space (i.e., relative to other stimuli) remains constant. Such stimulus-space stability could enable downstream cortical areas to read out an accurate perceptual representation in the face of drifting patterns of activity within V1.

## Discussion

We fit a computational model of V1 to a large longitudinal fMRI dataset, and measured cross-session model generalization using cvR² and correlation. We found that cvR² decreases with increasing intervals

between training and testing, while correlation does not, indicating that individual voxel response amplitudes gradually change across sessions. These changes are not uniform across V1, and result in progressively changing population responses. Relative similarities between population responses to multiple stimuli, however, remained stable throughout the study. Together, these findings provide evidence of representational drift in human primary visual cortex, and suggest a mechanism by which the visual system may overcome such changes.

What factors could underly changes in mean response amplitude in the absence of any overt training protocol? The small number of repeats and long periods of time between them make it unlikely that repetition effects such as fMRI adaptation or repetition suppression[7,11–13] were the root cause of the representational drift that we observed. It is also unlikely that changes in neurovascular coupling caused the drift, both because hemodynamic response functions (HRFs) have been shown to be stable over months[14], and because a change in the HRF would likely result in a multiplicative change in response gain, instead of an additive change in the baseline, as we report here. We consider three plausible mechanisms for the drift observed in our analysis. First, it is possible that drift is associated with changes in the neural responses caused by perceptual learning or mnemonic processes[15]. Subjects were performing a memory task that may have engaged neural circuits that exhibited priming. Although priming is most clearly evident in ventral visual cortical areas[16], priming effects may have resulted in feedback that altered responses in early visual cortex as well. Second, it is plausible that drift reflects changes in arousal. Performing a periodic task evokes widespread activity across visual cortex that is time-locked to the task[17–21], and this task-related activity is modulated by arousal[19,22]. As participants underwent repeated fMRI scanning week after week over many months, they may have habituated to the scanning environment, and their arousal may have gradually decreased over the course of the experiment. A systematic change in arousal may have resulted in additive changes in the amplitude of the task-related response[23], which could have resulted in drift. Indeed, changes in arousal could have been magnified by exogenous factors such as seasonality or changes in light exposure that in turn influence factors such as circadian rhythms[24,25]. Finally, a third possibility is that representational drift derives from stochastic changes in the neural responses. Random processes may lead to changes in neural population activity that occur at a range of timescales and may be cumulative in nature. In contrast to the first two explanations, under this scenario, the mean response amplitude was modulated by noisy, stochastic neural processes not directly related to the task or stimulus, but nonetheless resulting in changes to the neural responses to the stimuli. Longitudinal studies with concurrent physiological measures, such as pupil size and heart rate, may help to identify the different factors that may contribute to drift and differentiate between these three possibilities.

A number of previous studies have described representational drift in animal models. Here, we provide evidence for representational drift in human visual cortex. Aside from the species difference, our approach differs from previous studies in animals in several important aspects. First, previous studies typically presented the same series of stimuli (e.g. video frames) repeatedly within and across sessions[2,3]. This experimental design introduces a concern that the measurement of representational drift could be conflated by repetition effects (representation suppression or adaptation)[7]. In contrast to previous studies, here we analyze responses to a distinct set of stimuli presented in each session. While each of the 10,000 images was presented up to 3 times across the 30+ sessions of the entire experiment, the low frequency of repeats and the large number of images likely minimized repetition effects[11,12,26]. Second, because the NSD did not consist of responses to multiple repetitions of the same stimulus, we developed an analysis that differed from the analysis typically used in previous studies of

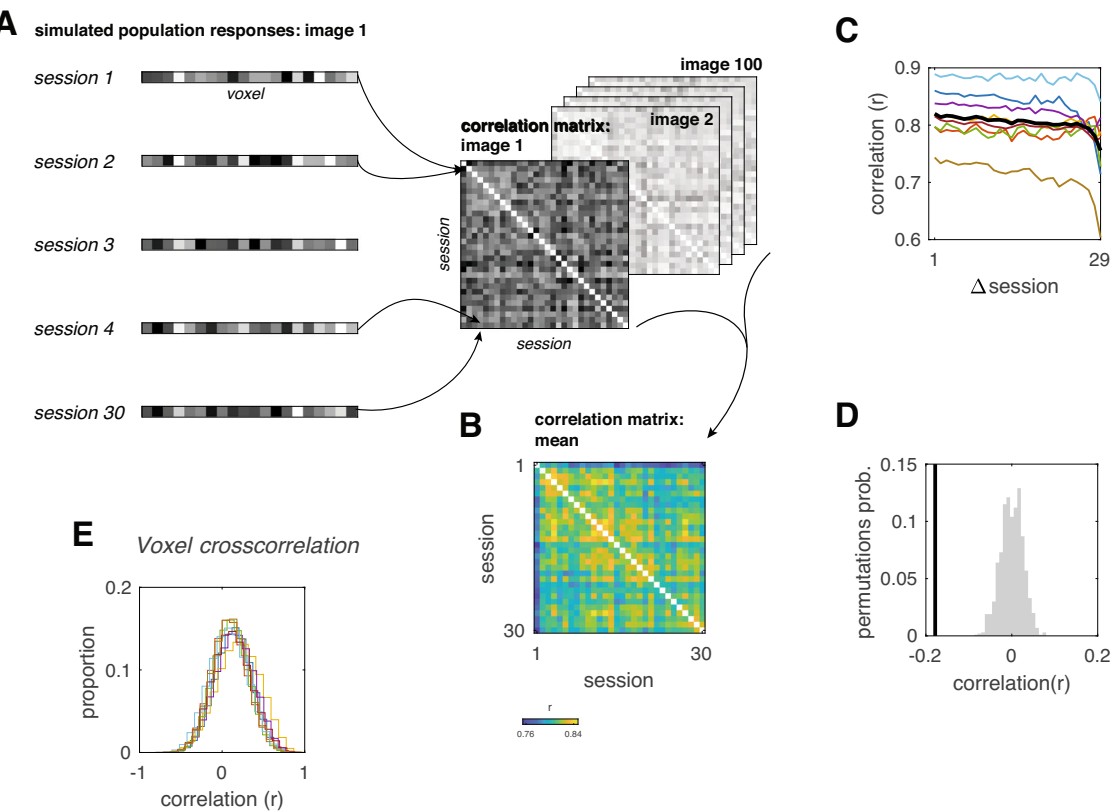

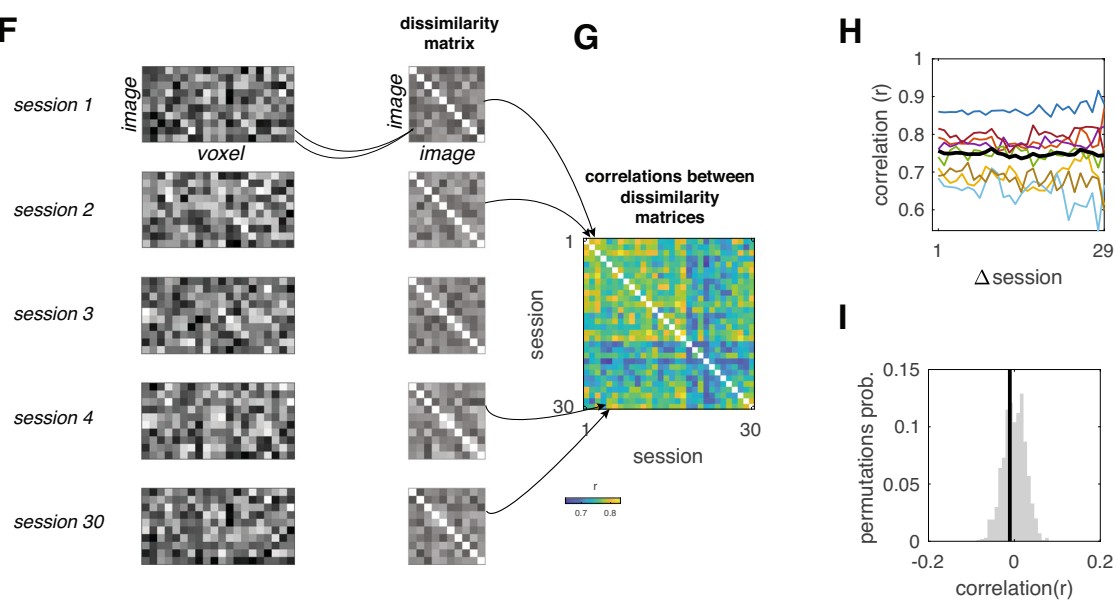

representational drift. Previous studies computed the correlation between responses to the same stimuli across sessions[2,3]. But because different images were presented in each NSD session, in order to compare responses to different stimuli we instead used an image-computable model that was based on a model that we and others have previously used to estimate the tuning characteristics of fMRI

measurements in V1. Thus, instead of comparing responses across sessions, we compared model weights across sessions. We then used the estimated tuning to simulate responses to the stimuli presented in the other scanning sessions. This approach enabled us to estimate responses to stimuli presented in different sessions, while minimizing repetition effects. One potential downside of our approach is that the

**Fig. 3 | Top, Population responses drift across sessions. A** Schematic illustration of analysis pipeline. Population responses to 100 images were simulated using the model weights estimated from each session. For each image, correlations were computed between all sessions, yielding a correlation matrix. These matrices were then averaged across all 100 images. **B** Empirical population response correlation matrix. **C** Correlation drops as function of number of intervening sessions, reflecting representational drift of population responses ($r = -0.18$, $p < 0.001$). **D** Null distribution of correlation values. Black vertical line, empirical correlation between population response correlations and number of intervening sessions. Gray histogram, null distribution of correlation values computed by randomizing order of sessions 1000 times. Bottom, representational dissimilarity matrices are stable across sessions. **E** Cross-correlation of voxels' mean response amplitude across sessions. Each colored line is the distribution of cross-correlations for V1 voxels from a single subject. Some correlations are positive while others are negative, indicating that voxels are not undergoing a uniform change in mean response amplitude across the entire V1. **F** Schematic illustration of analysis pipeline. Simulated population responses to different images were correlated with each other, yielding a dissimilarity matrix for each individual session. Next, correlations were computed between each possible pair of dissimilarity matrices. **G** Empirical correlation matrix. **H** Correlation between dissimilarity matrices does not drop with increasing number of intervening sessions, indicating stability across sessions ($r = -0.01$, $p = 0.339$). **I** Null distribution of correlation values. Black vertical line, empirical correlation between dissimilarity matrix correlations and number of intervening sessions. Gray histogram, null distribution of correlation values computed by randomizing the order of sessions 1000 times. Source data are provided as a Source Data file.

actual tuning will necessarily differ from the estimated tuning using the model because any noise in the measurement will interfere with the estimation procedure. But such inaccuracies should not accumulate across sessions. So, while error in the model fits will introduce noise, the use of the model cannot itself be the cause of drift. Third, while previous studies have measured representational drift using correlations, here we compared and contrasted correlation analysis with cross-validated $R^2$ in order to pinpoint the source of representational drift. Multiple analyses or statistical measures often converge to similar results, providing corroborating evidence[27,28]. On the other hand, slightly different analyses can probe different aspects of a phenomenon, shedding light on the underlying neural mechanism[29]. When measuring model goodness-of-fit with cvR$^2$, we found drift; but when measured with correlation, we did not. These two measures are sensitive to different aspects of the model prediction: while cvR$^2$ is sensitive to tuning shape, tuning gain and tuning baseline, Pearson's correlation is insensitive to changes in gain and baseline. Taking this difference into account we conducted additional analyses to pinpoint the drift as changes in the mean response amplitude. Thus, the two measures yielded complementary results, each providing unique information regarding the stability of neural tuning. We therefore believe that despite the differences, we are indeed measuring representational drift, as characterized previously in mouse visual cortex.

A recent study in mouse visual cortex reported changes in mean firing rates across sessions[2], consistent with our findings here. However, in contrast to our study, changes in tuning were observed, which seems to support a growing body of evidence on tuning changes during representational drift in rodent[1,3]. It is possible that the difference between our findings and these observations in rodents is related to a difference in the spatial scale of the measurement. Each fMRI voxel reflects the pooled activity of hundreds of thousands of neurons[30]. Changes that occur independently within each neuron are likely to cancel out in an fMRI measurement. On the other hand, population measurements will effectively amplify changes in neural activity that are correlated across nearby neurons[31,32]. It is possible that tuning changes occur independently for individual neurons, canceling out at the population level, as measured with fMRI[33]. But changes in mean response amplitude, on the other hand, may occur at a broader spatial extent, resulting in changes that are correlated across neighboring neurons, and resulting in a significant modulation in the voxel's mean response amplitude.

Previous studies in mouse visual cortex have demonstrated that representational drift tends to occur along dimensions that are not informative for downstream regions[2,4]. Similarly, we found that representations in human V1 change along dimensions that do not impact representational dissimilarity matrices. Such matrices quantify the relative representation between stimuli, which has been suggested to underlie behavior[34,35]. Stable relationships between individual percepts may enable downstream regions to guide consistent and stable behavior, despite changes that occur in upstream representations[36].

Further studies modeling representations throughout the visual hierarchy may help determine whether representational drift has a direct behavioral correlate.

During each trial of the NSD experiment, subjects fixated the center of a gray screen for 1 s, followed by the presentation of a single image for 3 s. While subjects were instructed to fixate during this 3 s stimulus presentation, it is likely that some subjects performed eye movements to scan the images. Yet our analysis pipeline assumed that subjects maintained central fixation, and that population receptive fields (RFs) sampled from a fixed location within the image. If subjects were to shift gaze and fixate on a region that is not the center of the image, the visual stimulation to each voxel would differ from the assumed location of the pRF in the image, and this would have the effect of adding noise to the model fits. It is possible that eye movement patterns change progressively across sessions, which would change the region in the image stimulating each voxel, and could in principle result in progressive changes in cross-session generalizability (i.e., drift). Eye tracking data included in the NSD dataset are insufficient to analyze either eye movements or pupil size in relation to drift. Concurrent high-quality eye tracking would be necessary to determine whether changes in eye movements are related to representational drift in visual cortex, as reported in mice[37]. The lack of eye tracking data also means that we cannot use pupil size measurement as a proxy for arousal. Recently a number of methods have been proposed for extracting arousal signals directly from fMRI data[38,39]. As these methods become more robust, it may be interesting to use a data-driven estimate of arousal to test whether changes in arousal correspond to representational drift.

We fit a model of spectral tuning (i.e., spatial frequency and orientation tuning), but assumed that spatial tuning (i.e., the pRF size, shape, and location) remains constant. However, it is possible that spatial tuning also changes over time. Since we have only a single independent measurement of the pRF, we assumed that the pRF remains stable, and asked whether spatial frequency and orientation tuning changed over time. If the pRF center or size were to change, we would expect that the spectral model goodness-of-fit tested on adjacent sessions would also change. But we found no evidence for changes in the spectral model goodness-of-fit. Since in the current study we have only a single measurement of the pRF, we cannot distinguish between changes in spatial and spectral tuning. An approach measuring both spatial and spectral tuning across sessions may be able to pinpoint which tuning aspects change over time.

The discovery of representational drift potentially challenges fundamental theories of neural coding, the simplest of which is the labeled line code. Here, we found that both stimulus tuning and gain are stable, while only response baseline changed over time. This type of change is consistent with the classical view of visual cortex activity as a labeled line code[40,41], but may pose a challenge for magnitude coding theories[42,43]. As noted above ('Spatial scale of representational drift'), however, this finding may depend on measurement granularity and spatial resolution. We further found that population responses to

single images change over time, but relative similarities between population responses remain stable, lending support to relational coding theories[44,45]. Further investigation of representational drift and other forms of tuning changes are needed to advance our understanding of the neural code.

# Methods

## Natural scenes dataset

The Natural Scenes Dataset (NSD; http://naturalscenesdataset.org)[8] contains measurements of fMRI responses from 8 participants who each viewed 9,000–10,000 distinct color natural scenes (22,000–30,000 trials) over the course of 30–40 scan sessions. Scanning was conducted at 7 T using whole-brain gradient-echo EPI at 1.8-mm resolution and 1.6-s repetition time. Images were taken from the Microsoft Common Objects in Context (COCO) database[46], square cropped, and presented at a size of 8.4° x 8.4°. A special set of 1,000 images were shared across subjects; the remaining images were mutually exclusive across subjects. Images were presented for 3 s with 1-s gaps in between images. Subjects fixated centrally and performed a long-term continuous recognition task on the images. The fMRI data were pre-processed by performing one temporal interpolation (to correct for slice time differences) and one spatial interpolation (to correct for head motion). A general linear model was then used to estimate single-trial beta weights. Cortical surface reconstructions were generated using FreeSurfer, and both volume- and surface-based versions of the beta weights were created.

In this study, we used the 1.8-mm volume preparation of the NSD data and version 3 of the NSD single-trial betas in percent signal change units (betas_fithrf_GLMdenoise_RR). Repeating the analyses using version 2 of the betas yielded similar results as those presented here. The results in this study are based on data from all NSD scan sessions, from all 8 subjects who participated in the NSD study. Since some subjects participated only in 30 sessions, we used the first 30 sessions for all subjects.

## Stimuli

NSD images were originally 425 × 425 pixels, and were then upsampled for display purposes to 714 × 714 pixels. We reproduced this upsampling in our stimulus preparation, and padded the images with a gray border on all four sides (mimicking the scanner display environment), resulting in a final image dimension of 1024 × 1024 pixels. A semitransparent red fixation point was added at the center to simulate the actual stimulation experienced by the subjects during the experiment. Images were converted to grayscale by averaging across the 3 color channels. To speed up subsequent computations, the images were then downsampled to 512 × 512 pixels.

## Steerable pyramid

The image-computable model, based on the steerable pyramid, simulates each neuron in V1 with a receptive field that is tuned for both spatial frequency and orientation, and then allows for variable weighting of these model neurons. It is possible to create steerable pyramid models with a wide range of parameters, each instantiating different hypotheses regarding the tuning properties of individual neurons. We used a steerable pyramid with 8 orientations, 7 spatial frequency levels, and a spatial frequency bandwidth of 1 octave, resulting in tuning profiles that resemble those of individual V1 neurons[47]. In addition to oriented filters, we included 2 filters for the lowest and highest spatial frequencies, yielding a total of 58 filters. In contrast to energy models we used in previous studies[32,48], here we took the square root of the filter outputs, which resulted in slightly better model fits. Using only the 56 oriented filters and squaring the outputs yielded similar results, indicating that our findings are robust to model specifics.

## pRF modeling

pRF estimates are included in the NSD[8]. Briefly, pRFs were estimated based on a single session (6 runs, 300 s each) of a pRF mapping experiment. Stimuli consisted of slowly moving apertures (bars, wedges, and rings) filled with a dynamic colorful texture, that appeared within a circular region of 8.4 deg diameter. Subjects performed a color change detection task at fixation. pRFs were estimated using the Compressive Spatial Summation (CSS) model[49].

## Regions of interest

Regions of interest V1, V2, V3, hV4 were defined in the NSD dataset based on the pRF maps. In this study we analyzed all 4 regions but focused on V1 where orientation selectivity has been studied most extensively. Results are presented for V1 only, including all voxels with pRF $R^2$ >0.

## pRF sampling

The output of each filter in the steerable pyramid was sampled by each voxel's pRF by multiplying the 2D pRF with the filter output. The pRF was modeled as a 2D isotropic (circular) Gaussian, using the 'size' parameter as the Gaussian's standard deviation. (Note that the 'size' parameter, as estimated as part of NSD, reflects the response of the modeled pRF to point stimuli and takes into account the exponent used in the CSS model.) For filter $k$ of image $j$ ($\mathbf{F^{j,k}}$), the sampled output for voxel $i$ with a pRF centered at $(x_i, y_i)$ and standard deviation of $\sigma_i$, is computed as dot product between the pRF and the filter:

$$f_i^{j,k} = \sum_{x,y} \mathbf{F_{x,y}^{j,k}} \cdot e^{-\frac{(x_i-x)^2 + (y_i-y)^2}{2\sigma_i^2}} \tag{1}$$

The model had 58 sampled outputs per image, for each voxel.

## Normalizing variance and mean response

To remove changes in the mean response amplitude across sessions, we subtracted the mean beta from each session. To remove changes in variability across sessions, we first subtracted the mean, then z-scored the session, and then added back the original mean. This resulted in a STD of 1, while keeping the mean unchanged. To remove changes in the V1 mean response, we subtracted the V1 mean response in each session from all voxels.

## Multiple regression

We modeled the responses of voxel $i$, $\mathbf{y_i}$, as a linear combination of the sampled filter outputs and a constant term plus noise:

$$\mathbf{y_i} = \mathbf{f_i} \cdot \boldsymbol{\beta_i} + \boldsymbol{\varepsilon_i} \tag{2}$$

Here $\mathbf{f_i}$ is a matrix consisting of voxel $i$'s sampled outputs for all filters of all images and a constant term (images x filters+1). $\boldsymbol{\beta_i}$ is a vector of beta weights (filters+1 ×1), and $\boldsymbol{\varepsilon_i}$ is a set of residuals (images x 1).

Beta weights were estimated using ordinary least-squares:

$$\hat{\boldsymbol{\beta_i}} = (\mathbf{f_i^T f_i})^{-1} \mathbf{f_i^T y_i} \tag{3}$$

Note that each voxel not only had different beta weights but also different predictors due to the incorporation of each voxel's unique pRF, thus distinguishing this regression from a general linear model analysis of the voxel responses.

## Goodness-of-fit measures

To assess model goodness-of-fit, we performed cross-validation. After estimating model parameters on session $j$, the regression

prediction was calculated as:

$$\widetilde{\mathbf{y}}_\mathbf{i}^\mathbf{pred} = \mathbf{f}_\mathbf{i} \cdot \widetilde{\boldsymbol{\beta}}_\mathbf{i} \tag{4}$$

where $\mathbf{f}_\mathbf{i}$ is constructed for session $k$, and $\widetilde{\boldsymbol{\beta}}_\mathbf{i}$ are the betas weights estimated using session $j$. The residual of this prediction is given by

$$\widetilde{\mathbf{y}}_\mathbf{i}^\mathbf{resid} = \mathbf{y}_\mathbf{i} - \widetilde{\mathbf{y}}_\mathbf{i}^\mathbf{pred} = \mathbf{y}_\mathbf{i} - \mathbf{f}_\mathbf{i} \cdot \widetilde{\boldsymbol{\beta}}_\mathbf{i} \tag{5}$$

Cross-validated $R^2$ is then computed as

$$cvR_i^2 = 1 - \frac{SS\left(\widetilde{\mathbf{y}}_\mathbf{i}^\mathbf{resid}\right)}{SS(\mathbf{y}_\mathbf{i} - \bar{y}_i)} \tag{6}$$

where $\bar{y}_i$ is the mean response across images, and $SS$ denotes the sum of squares.

When performing a regression analysis, $R^2$ values by definition are within the range of 0 and 1. In that context, a negative $R^2$ would indicate that something is wrong with the calculation. However, here we are using cross-validated $R^2$ (cv$R^2$), which is not constrained to the range of 0 and 1. For cv$R^2$, the difference between positive and negative values is quantitative, not qualitative. For example, predicting the mean of the test data, would result in cv$R^2 = 0$. This may seem like a low value, but actually may indicate a model with good predictive power. This is because the mean is not provided to the model, but rather, is accurately predicted in the left-out data. Importantly, a negative cv$R^2$ value does not indicate something wrong with the model or the calculation. Rather, it indicates relatively low prediction accuracy, which could be due to limited training data. Critically, what is important for the current study is the change in cv$R^2$, rather than the absolute value, since decreasing values indicate that something about the predictive power of the model is changing with time.

In addition to $cvR^2$, goodness-of-fit was quantified by Pearson's correlation between the prediction and the measured responses.

Note that $R^2$ is sensitive to changes in baseline and gain, while correlation is not. In other words, if the prediction is a scaled version of the recorded responses ($\widetilde{\mathbf{y}}_\mathbf{i}^\mathbf{pred} = A\mathbf{y}_\mathbf{i}; A \neq 1; A > 0$), or if they differ in the baseline value ($\widetilde{\mathbf{y}}_\mathbf{i}^\mathbf{pred} = \mathbf{y}_\mathbf{i} + B; B \neq 0$), the $R^2$ value will be <1, while Pearson's correlation will equal 1.

Each entry in the goodness-of-fit matrix for a single subject consisted of the median across all V1 voxels.

To visualize how goodness-of-fit measures depend on the number of intervening sessions between training and testing, we computed the mean across all goodness-of-fit matrix entries that correspond to the same interval. This corresponds to averaging matrix diagonals, where the distance from the main diagonal reflects the number of intervening sessions.

The number of sessions each subject was scanned ranged from 30 to 40. The goodness-of-fit matrix used all subjects' first 30 sessions, so that results could be averaged across all subjects. Similarly, goodness-of-fit as function of intervening sessions was computed for all intervals up to a maximum of 29 intervening sessions.

To measure the correspondence between goodness-of-fit measures and number of intervening sessions, we computed the correlation between goodness-of-fit and number of intervening sessions. Note that the samples used for the correlation are not independent. Each datapoint used in the correlation is the goodness-of-fit of a model trained on session $i$ and tested on session $j$. All combinations of $i$ and $j$ are included in the correlation analysis. Therefore, 29 datapoints use the same session for training, and 29 datapoints use the same session for testing. For this reason, we used a permutation procedure to test for statistical significance instead of a $t$-test which assumes independence between samples.

## Permutation test of significance
To determine whether the drift was statistically significant we used a permutation test. First, we quantified the amount of drift as Pearson's correlation between goodness-of-fit and number of intervening sessions for each entry in the goodness-of-fit matrix for each individual subject, and then averaged the correlation across subjects. Next, we determined whether the correlation is significantly lower than zero. Standard parametric and nonparametric tests are inappropriate in this case, since values for different numbers of intervening sessions are not independent. Instead, we generated a null distribution using a permutation test. We permuted the order of sessions in the goodness-of-fit matrix 1000 times (using the same permutation for all subjects) and recomputed the resulting correlation coefficient for each permutation. The $p$-value was computed as the proportion of permutations that yielded a correlation coefficient not greater than the empirical correlation. Note that permuting the session order is expected to cause the autocorrelation (e.g. of mean voxel response amplitudes) to be negative, on average. This is because for short timeseries the autocorrelation is negatively biased[50].

## Tuning change simulation
To understand how changing a voxel's tuning curve affects the mean and STD of the voxel's responses, we simulated a simple 1D gaussian tuning curve with 4 parameters:

$$R(x) = \frac{A}{\sqrt{2\pi\sigma^2}} e^{\frac{-(x-x_0)^2}{2\sigma^2}} + C \tag{7}$$

The baseline is $C$, gain is determined by $A$, tuning width is determined by $\sigma$, and the preferred stimulus is $x_0$. For different values of each parameter we computed the mean and standard deviation of the tuning curve to understand how both values are impacted by the tuning curve parameters.

## Simulating population responses
Since each image was presented a total of 3 times across all sessions, we could not systematically compare population responses across sessions. Instead, we used the model fits from each session to simulate responses to 100 randomly selected images. For each image we created a correlation matrix, consisting of correlations between the simulated population responses of each session and those of all other sessions. Next, we computed the mean of all 100 single-image matrices, yielding a single matrix per subject. From this single matrix we averaged all entries corresponding to each interval.

## Dissimilarity matrices (RDMs)
To test whether relative similarity between population responses to images is stable, we constructed a representational similarity matrix (RDM) using the simulated population responses for each session. Each RDM consisted of distances between all pairs of simulated population responses within a single session, computed as $1 - r$, where $r$ is Pearson's correlation coefficient. We then computed the Spearman correlation between pairs of RDMs, using the matrix values that are above the main diagonal (i.e. the upper triangle of the matrix).

## Reporting summary
Further information on research design is available in the Nature Portfolio Reporting Summary linked to this article.

## Data availability
The NSD dataset is freely available at http://naturalscenesdataset.org. Images used for NSD were taken from the Common Objects in Context database (https://cocodataset.org). Source data are provided with this paper.

## Code availability

Code for analyzing the data and generating the figures is available at: https://github.com/elimerriam/repDriftNSD[51].

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

## Acknowledgements

Special thanks to Corey Ziemba, Chris Baker, Charlie Burlingham, Robbe Goris, and Matthias Nau and members of the Laboratory of Brain and Cognition (LBC) at NIMH for helpful comments. This research was supported by the Intramural Research Program of the NIMH (ZIAMH002966) to E.P.M.

## Author contributions

Project administration, writing—review and editing: Z.N.R and E.P.M. Conceptualization, methodology, investigation, software, data curation, formal analysis, writing—original draft, visualization: Z.N.R. Resources, supervision, funding acquisition: E.P.M

## Funding

## Competing interests

The authors declare no competing interests.
