## [Peer Review File · Nature Communications]

Representations in human primary visual cortex drift over timeREVIEWER COMMENTS

Reviewer #1 (Remarks to the Author):

Roth & Merriam report on their finding of representational drift in human visual fMRI responses. They support their findings with detailed model-based analyses, and show that the drift they find likely results from changes in neural responsivity. The research is evidently expertly conducted. The writing and presentation suffer from a strong dedication to brevity, I think the clarity of the work can easily be ameliorated by expanding the narrative.

Points on writing and analysis:

In the introduction, the authors do not explain the interest neuroscientists should have in representational drift, and the repercussions it has for labeled-line coding. It remains relatively unclear why we should be interested in representational drift, which is a shame.

Perhaps the authors could explain a bit why negative coefficient of determination values are not problematic, as naive readers may come to this conclusion.

Although I agree with their choice to limit their analysis to 30 sessions, I do wonder what would happen if they were to conduct their analysis on the entire dataset for each subject. Is that single subject with the non-significant trend one of the 30 session subjects?

What is that band in figure 1F, around session 25? It seems to reappear also in figures 2B & 2G. What is the impact of this drop on the results, and how is this distributed across the subjects?

It would be good if the authors could argue convincingly why using simulated population responses instead of actual data doesn't stand in the way of their conclusions. For instance, could they not construct RDMs from repeated images across sessions, and would this not strengthen their findings?

In 2H, should the authors not be performing a Bayesian analysis, instead of arguing for the non-existence of an effect from a non-significant p-value?

How robust are the findings to changes in pRF location and size, that may result from the difference in task between pRF mapping and the NSD core experiment — which featured a mnemonic task?

Slight textual points:

line 99: double *is*

line 141: duration => number

Reviewer #2 (Remarks to the Author):

In this manuscript, Roth & Merriam examine whether the human visual cortex exhibits representational drift. This is an interesting topic that has received a lot of interest in animal models, but has not been investigated in humans. The authors use a large Natural Scenes Dataset (NSD) to investigate whether drift might occur across sessions. I did have some concerns that the NSD is not ideal for analyzing representational drift due to the lack of repeated images. The authors get around this in part by looking at the drift in model fits (essentially voxel tuning) across sessions. This is clever, but is a little different than representational drift as typically described – i.e., changes in response to the same stimulus. I have a number of additional concerns about the analysis approach and interpretation, detailed below.

Specific concerns:

1. One point I found confusing is that the authors are not investigating the change in response to the same stimuli, as has typically been done in the representational drift literature. I assume this is due to the low number of repeats (3 total across sessions) in the NSD. I like their approach for getting around that limitation (looking at modeled vs actual responses to the stimuli in each session), but they are really looking at the stability of the model fits, not the stability of the responses, which is a different way of looking at drift. They should make this point clearly in their text.

2. Related to the first point, it would be nice if they did look at drift in responses in the same stimuli when they are repeated across sessions. Given, there would be more noise, and each stimulus would only have a few `_delta_session` intervals, but averaging across the large database of images should yield reliable results.

3. For all of the drift analyses, the difference in session number was used as the dependent variable. However, I suspect the difference in time (days) between sessions is likely the more relevant variable in terms of representational drift.

4. The NSD has eye tracking data. This would be useful for excluding trials with major eye movements (maybe this was done already, but I couldn't find it in the methods) and looking for progressive changes in behavioral state (increased # of eye movements, changes in pupil dilation, etc). Recent work from the Clopath lab has suggested that some changes in apparent representational drift can be explained by behavioral state changes (at least over short time intervals), so it would be preferable to control for this variable given that the data is available.

5. One major point of confusion for me was the poor fit of their models (i.e., $R^2 < 0$). I believe the negative values arise from their cross-validation procedure (since typically $0 < R^2 < 1$). In any case, this indicates that their model fits are doing a very poor job of predicting the data. I suspect this may happen in part due to the low threshold for including voxels (pRF $R^2 > 0$). I would encourage the authors to be more stringent about including only visually-driven voxels (perhaps derive a null distribution for pRF R^2 and only keep significant voxels?). This may improve the model fits and interpretability of the data. As it is, I find it hard to make sense of the decrease in negative R^2 over time. It also concerns me that the result did not hold up using the Pearson correlation measure, which is more similar to other measures used in the animal model literature.

We thank the reviewers for their insightful comments and suggestions regarding the revision of our manuscript. Both reviewers raised a number of important issues. These helpful comments have prompted a significant revision of the text, which we feel has greatly improved the manuscript and which we hope addresses the reviewers' concerns. Below, we provided a point-by-point response to each of their comments.

REVIEWER COMMENTS

Reviewer #1 (Remarks to the Author):

Roth & Merriam report on their finding of representational drift in human visual fMRI responses. They support their findings with detailed model-based analyses, and show that the drift they find likely results from changes in neural responsivity. The research is evidently expertly conducted. The writing and presentation suffer from a strong dedication to brevity, I think the clarity of the work can easily be ameliorated by expanding the narrative.

We thank the reviewer for the positive assessment of our work and for the helpful comments. Indeed, the manuscript was originally written to target a brief format publication. In the revision, we have expanded the manuscript considerably, as detailed below, and believe it is now much clearer and easier to understand.

Points on writing and analysis:

In the introduction, the authors do not explain the interest neuroscientists should have in representational drift, and the repercussions it has for labeled-line coding. It remains relatively unclear why we should be interested in representational drift, which is a shame.

As the reviewer points out, representational drift could potentially be problematic for labelled lines and other theories of neural coding. It is classically assumed that visual stimuli are coded by labeled lines (Barlow 1972, Marr 1982). The labeled line coding theory has been challenged by the demonstration that various experimental manipulations can affect neural tuning. In a similar vein, representational drift may constitute a challenge to the labeled line theory because representational drift may reflect changes in neural tuning. However, we found that drift was primarily driven by changes in responses gain (not a change in selectivity), and a change in response gain is consistent with labeled line coding (e.g., David et al., Neuron, 2008). To address the reviewer's comments we have expanded the Introduction to include a broader framing of why representational drift is important for neural coding, and we have added a section to the Discussion dealing with the implications for labeled lines and neural coding more generally.

Perhaps the authors could explain a bit why negative coefficient of determination values are not problematic, as naive readers may come to this conclusion.

This is an important point that we have tried to make much clearer in the revision. We agree that negative R^2 values seem unusual, but we believe they are sensible given the particular analysis that we have performed here, and we have added a section to the Methods ("Goodness-of-fit measures") explaining why.

When performing a regression analysis, R^2 values by definition are within the range of 0 and 1. In that context, a negative R^2 would indicate that something is very wrong with the calculation. However, in our study, we are using cross-validated R^2 (cvR^2), which is not constrained to the

range of 0 and 1. For cvR^2 , the difference between positive and negative values is quantitative, not qualitative. For example, predicting the mean of the test data would result in $cvR^2=0$. This may seem like a low value, but actually may indicate a model with good predictive power. This is because the mean is not provided to the model, but rather, is accurately predicted by the left-out data.

Importantly, a negative cvR^2 value does not indicate something wrong with the model or the calculation, Rather, it indicates relatively low prediction accuracy, which could be due to limited training data. We believe that the primary limiting factor in our case is the amount of data used to train the model. In our previous study (Roth et al., 2022), we used half of each subject's data to train the model, and as a result the vast majority of cvR^2 values were positive. In the current study, we used a single session to train the model, which is not a lot of data. A secondary factor contributing to negative R^2 values may be the difference in baseline response amplitude that changes across sessions. When we normalize mean response amplitude (Fig 2, bottom), the cvR^2 values are higher, yet still negative.

Critically, what is important for our study is the change in cvR^2 , rather than the absolute value, since decreasing values indicate that something about the predictive power of the model is changing ('drifting') with time. We have now used the term cvR^2 , instead of R^2 , to emphasize the difference between the two calculations, and we hope that our expanded discussion of this point in the Methods section makes this point clearer.

Although I agree with their choice to limit their analysis to 30 sessions, I do wonder what would happen if they were to conduct their analysis on the entire dataset for each subject. Is that single subject with the non-significant trend one of the 30 session subjects?

Our report in the initial submission that one subject was non-significant turns out to have been a technical error. When reviewing the data we found that drift was significant in all subjects ($p<0.05$). Following the reviewer's suggestion, we reran the analysis using all sessions for each subject (see below), and found that the results were largely the same.

Figure R1: Analysis using all sessions for each subject. cvR^2 (left) and Pearson's r (right) as function of number of intervening sessions between training and test sessions.

What is that band in figure 1F, around session 25? It seems to reappear also in figures 2B & 2G. What is the impact of this drop on the results, and how is this distributed across the subjects?

The band around session 25 means that those sessions generalize less well to other sessions. Despite examining the data extensively, we do not have an explanation for why there is a drop at session 25. Interestingly, this band is absent from the cvR^2 matrix. To test the impact of this band on the results, we repeated the analysis using only the first 20 sessions, and found drift in

all subjects, indicating that this band is not the cause of the drift findings or a source of a false positives in the results.

Figure R2: Analysis limited to first 20 sessions. Top, cvR^2 ; Bottom, Pearson's r . Left: goodness-of-fit matrix. Center: goodness-of-fit as function of number of intervening sessions between training and test sessions. Colored thin lines, individual subjects; thick black line, mean across subjects. Right: null distribution of correlation values (gray histogram), and the empirical value (black vertical line).

Looking at each individual subject, it seems that this band is most prominently evident in S2 and S3. Single subject goodness-of-fit matrices (cvR^2 on top, Pearson's correlation on bottom):

Figure R3: cvR^2 (top) and correlation (bottom) matrices for individual subjects

It would be good if the authors could argue convincingly why using simulated population responses instead of actual data doesn't stand in the way of their conclusions. For instance, could they not construct RDMs from repeated images across sessions, and would this not strengthen their findings?

The reviewer asks whether we can construct RDMs from repeated images across sessions. Unfortunately, the structure of the NSD dataset precludes this particular analysis. Each image was presented a total of 3 times throughout the 40 sessions. In each session, a different subset of images was presented. For each image, the 3 repetitions are in a different set of sessions, and for subjects who participated in less than 40 sessions, many of the images were presented only once or twice.

We searched for subsets of images that were presented together in the same 3 sessions (and up to session 30). The largest such set consists of 5 images, and there was only 1 such set. Such an RDM is based on a total of 5 trials, and is very noisy, so we cannot determine whether changes in the RDM across the 3 sessions are the result of noise or of drift. Therefore, we were

unable to construct RDMs using measured responses to the same images across different sessions, as the reviewer requested.

Figure R4: number of trials consisting of 1st, 2nd, and 3rd repetition in each session.

While it was not possible to test for drift using RDMs from measured responses, simulating such RDMs with the model was a viable alternative. The analysis we ran estimated how RDMs evolve over time. As with the model generalization analysis (Fig 1), we rely on the model to extract each voxel's tuning. This tuning allowed us to estimate responses to any arbitrary stimulus. The simulated RDMs won't be identical to RDMs we would get by measuring responses to the same set of images repeated over time, but it is an approximation up to effects of noise that reduce estimation accuracy (both in the measured responses, and in the model fits). The crucial point is that, while these inaccuracies may fluctuate, the fluctuations should not accumulate over time, producing the appearance of artifactual drift. If relative similarities between representations are stable, then the simulated RDMs should be stable. If, on the other hand, relative similarities between responses to the same images change over time, then these changes should be reflected in changes in the model weights and should thus be captured in the simulated RDMs. In other words, using the model weights instead of measured RDMs may have added random noise in the analysis, but it could not have produced spurious drift.

In 2H, should the authors not be performing a Bayesian analysis, instead of arguing for the non-existence of an effect from a non-significant p-value?

In the version of the analysis reported in the initial submission, this analysis yielded a p-value of 0.059. Using an improved version of the model with additional pyramid filters, as described in detail in the resubmission, we find the RDMs to be much more stable (Fig 2, bottom). The lack of correlation with the number of intervening sessions is unambiguous ($r=-0.01$) and the p-value clearly not significant (0.34). We hope this reanalysis of the data allays the reviewer's concerns.

As a general point, we take to heart the reviewer's concerns that we cannot accept the null hypothesis that there is no effect, and a Bayesian approach is often more appropriate given this point. But an analysis based on Bayes factors would not be possible given the structure of these data because the samples are not independent. Each datapoint used in the correlation is the goodness-of-fit of a model trained on session i and tested on session j (we now stress this point in the Methods). All combinations of i and j are included in the correlation analysis. Therefore, 29 datapoints use the same session for training, and 29 datapoints use the same session for testing. This dependence is the reason we used a permutation procedure to test for significance instead of a t-test, which assumes independence between samples. Following discussions with

several statisticians at our institution (NIH), our understanding is that there is currently no straightforward method to run Bayes factor analysis in this situation.

How robust are the findings to changes in pRF location and size, that may result from the difference in task between pRF mapping and the NSD core experiment — which featured a mnemonic task?

As the reviewer states, task type could possibly affect the measured pRF. pRFs were estimated in a separate pRF mapping session with the traditional traveling bar stimulus and a concurrent color-detection task at fixation. The main NSD experiment involved a memory task. Perhaps this task difference caused a mismatch between the estimated pRFs and the actual pRFs during the NSD experiment?

In order to address this question, the original NSD study estimated pRFs both from the pRF mapping session, and from the NSD core experiment (Allen et al. 2021, Extended Data Fig 9). They found very similar pRF estimates using both datasets, with no systematic differences between the sets of pRF estimates. Furthermore, if the pRF mapping task had biased the pRF estimate to differ from the actual spatial tuning in the NSD experiment, this would result in poorer model fits. However, this bias would be constant, and should not introduce nor extinguish any spectral tuning changes over time.

In the representational drift analysis, we fit a model of spectral tuning (i.e. spatial frequency and orientation tuning), but assumed that spatial tuning (i.e. the pRF) remains constant. It is conceivable that spatial tuning changes over time, but since we have only a single measurement of the pRF, we assumed that the pRF remains stable, and asked instead whether spatial frequency and orientation tuning changed over time. If the pRF center or size were to change, we would expect that the spectral model goodness-of-fit tested on adjacent sessions would change over time. But we found this not to be the case, implying that pRFs are stable. Since in the current study we have only a single measurement of the pRF, we cannot distinguish between changes in spatial and spectral tuning. Distinguishing between these two possibilities in future studies would require measuring both spatial and spectral tuning across sessions in order to pinpoint which tuning aspects change over time.

Slight textual points:

*line 99: double *is**

line 141: duration => number

These errors have been fixed. We thank the Reviewer for pointing them out.

Reviewer #2 (Remarks to the Author):

In this manuscript, Roth & Merriam examine whether the human visual cortex exhibits representational drift. This is an interesting topic that has received a lot of interest in animal models, but has not been investigated in humans. The authors use a large Natural Scenes Dataset (NSD) to investigate whether drift might occur across sessions. I did have some concerns that the NSD is not ideal for analyzing representational drift due to the lack of repeated images. The authors get around this in part by looking at the drift in model fits (essentially voxel tuning) across sessions. This is clever, but is a little different than representational drift as typically described – i.e., changes in response to the same stimulus. I have a number of additional concerns about the analysis approach and interpretation, detailed

below.

Specific concerns:

1. One point I found confusing is that the authors are not investigating the change in response to the same stimuli, as has typically been done in the representational drift literature. I assume this is due to the low number of repeats (3 total across sessions) in the NSD. I like their approach for getting around that limitation (looking at modeled vs actual responses to the stimuli in each session), but they are really looking at the stability of the model fits, not the stability of the responses, which is a different way of looking at drift. They should make this point clearly in their text.

The reviewer correctly points out that our approach is different from that employed in previous investigations of drift in animal studies. This was because in the NSD experiment, each stimulus was repeated only 3 times over the course of many months, whereas in the animal studies the same stimulus was typically presented many times. We feel that there are pros and cons to each approach. Following the reviewer's suggestion, we have added a new section to the discussion addressing this issue and highlighting important differences between using measured responses to repeated presentation, vs. our approach that relied on model fits of responses to novel stimuli. In this new section, we highlight the pros and cons of each approach and argue that both should in theory be targeting the same underlying phenomenon (see Discussion: Relation to previous studies of representational drift).

2. Related to the first point, it would be nice if they did look at drift in responses in the same stimuli when they are repeated across sessions. Given, there would be more noise, and each stimulus would only have a few Δ session intervals, but averaging across the large database of images should yield reliable results.

Following the reviewer's suggestion, we performed an additional analysis that is more directly analogous to those in previous animal studies. For example, most analyses in previous representational drift studies in animal visual cortex involve correlating responses to a series of stimuli (e.g. a series of video frames) across sessions (e.g. McMahan et al. 2014, Marks & Goard 2021, Deitch et al. 2021 [analysis number 3]). Here, we cannot perform that analysis since different stimuli were presented in each session. Hence, we performed an alternative version of this analysis focusing on stimulus repetitions alone.

Limiting the analysis to repetitions of the same stimuli, we found that mean V1 response amplitude decreases with repetition number (repetition 3 < repetition 1, $p=0.026$). Additionally, we found that population responses become less similar with repetition number ($1^{\text{st}}-3^{\text{rd}}$ correlation < $2^{\text{nd}}-3^{\text{rd}}$ correlation, $p=0.012$; $1^{\text{st}}-3^{\text{rd}}$ correlation < $1^{\text{st}}-2^{\text{nd}}$ correlation, $p<0.0001$). Both of these findings are consistent with our report in the manuscript of representational drift estimated using the model. However, both of these findings are also consistent with repetition suppression or other forms of adaptation. Repetition suppression means that additional repetitions evoke smaller amplitude responses, which could explain the first finding. If different voxels exhibit differing degrees of suppression, this would result also in increasingly differing population responses, causing the second finding. Because of the susceptibility of this approach to adaptation and/or repetition suppression we feel strongly that the model-based approach reported in the manuscript is more appropriate for studying representational drift, given this particular dataset.

Figure R5: Left: mean beta weight as function of repetition number. Right: Mean correlation between repetitions, for each of 3 possible pairs of repetitions.

We next analyzed the 3 repetitions in each session separately. Representational drift was still evident in the trend of each repetition. However, because we divided the data into 3 sets we lacked statistical power and the results were not conclusive.

Figure R6: Left: mean beta weight as function of session number, for each repetition. Right: Correlation between repetitions as a function of the number of sessions between the 2 repetitions.

3. For all of the drift analyses, the difference in session number was used as the dependent variable. However, I suspect the difference in time (days) between sessions is likely the more relevant variable in terms of representational drift.

Figure R7: Left: cvR^2 as function of the number of intervening sessions between training and test sessions (Δ_{session}). Each point represents a pair of training and testing sessions for a single subject. Center: cvR^2 as function of the number of intervening days between training and test sessions. Right: session day as a function of session number. Color in all panels reflects subject identity, as in Fig 1D.

The reviewer raises an important question: Is representational drift a process that occurs at a fixed temporal rate, or alternatively is the rate of drift dependent on visual or task experience? In the NSD experiment, sessions were generally spaced 1 week apart. Any changes that correlate with the number of intervening sessions should therefore also correlate with the number of intervening days. Indeed, we found that both correlations were equal ($r=-0.17$). We therefore cannot determine whether the rate of drift depends on the amount of time or on the number of intervening sessions. In order to address this question, it would be interesting in a future study to schedule sessions with either a fixed periodicity or non-periodically (e.g., a randomly-selected day of the week), to distinguish between these two factors, and determine whether drift is driven by time or by experience.

4. The NSD has eye tracking data. This would be useful for excluding trials with major eye movements (maybe this was done already, but I couldn't find it in the methods) and looking for progressive changes in behavioral state (increased # of eye movements, changes in pupil dilation, etc). Recent work from the Clopath lab has suggested that some changes in apparent representational drift can be explained by behavioral state changes (at least over short time intervals), so it would be preferable to control for this variable given that the data is available.

We agree with the reviewer that it is conceivable that interesting features of representational drift could be revealed by analyzing saccade frequency or other ocular factors over the course of the year-long experiment. However, the eye tracking data provided by the NSD dataset is insufficient for this purpose. Only a small number of NSD sessions included eye tracking. Between 2 and 4 eye tracking sessions were collected per subject, including 2 sessions that were not part of the core NSD experiment (i.e., one 'imagery' session and one 'synthetic' session with grating stimuli). In other words, between 0 and 2 sessions of the core NSD experiment included eye tracking. Two subjects had no eye data collected during the core NSD experiment. The data collected for the other 6 subjects was of inconsistent quality (see Allen et al. 2021, Extended Data Fig 4, panel C – numbers of good runs include runs from the imagery session and from the synthetic session). Therefore, the eye tracking data is insufficient to analyze either eye movements or pupil size in relation to drift.

We have added the following text to the Discussion, to make clear the implication of unaccounted for eye movements during the experiment:

During each trial, a single image was presented for 3 seconds. While subjects were instructed to fixate, it is likely that some subjects performed eye movements to scan the images. Yet the model-fitting assumed fixation at the image center. When subjects fixate on a region that is not the center of the image, the visual stimulation to each voxel will differ from the assumed location of the pRF in the image, adding noise to the model fits. It is possible that eye movement patterns change progressively across sessions, which would change the region in the image stimulating each voxel. Eye tracking data included in the NSD dataset is insufficient to analyze either eye movements or pupil size in relation to drift. Concurrent high-quality eye tracking would be necessary to determine whether changes in eye movements are related to representational drift in visual cortex, as reported in mice (Sadeh & Clopath, 2022, eLife).

The lack of eye tracking data also means that we cannot use pupil size measurement as a proxy for arousal. Recently a number of methods have been proposed for extracting arousal signals directly from fMRI data (Gonzales-Castillo et al., Neuroimage, 2022; Chang et al., 2016, PNAS). As these methods become more robust, it may be

interesting to use a data-driven estimate of arousal to test whether changes in arousal correspond to representational drift.

5. One major point of confusion for me was the poor fit of their models (i.e., $R^2 < 0$). I believe the negative values arise from their cross-validation procedure (since typically $0 < R^2 < 1$). In any case, this indicates that their model fits are doing a very poor job of predicting the data. I suspect this may happen in part due to the low threshold for including voxels ($pRF R^2 > 0$). I would encourage the authors to be more stringent about including only visually-driven voxels (perhaps derive a null distribution for $pRF R^2$ and only keep significant voxels?). This may improve the model fits and interpretability of the data. As it is, I find it hard to make sense of the decrease in negative R^2 over time. It also concerns me that the result did not hold up using the Pearson correlation measure, which is more similar to other measures used in the animal model literature.

Regarding negative cvR^2 values, we understand the reviewer's point. Reviewer 1 had a similar concern that we have addressed above. In short, R^2 values lie between 0 and 1 when R^2 is used as a measure of a model's goodness-of-fit. However, in our experiment we are using cvR^2 to quantify the goodness-of-fit across sessions (i.e., fitting and evaluating the model on independent data). We entertained alternate goodness-of-fit metrics that had positive values, but felt that cvR^2 was the most easily interpretable of all the alternatives.

While previous studies have measured representational drift using correlations, here we compared and contrasted correlation analysis with cross-validated R^2 in order to pinpoint the source of representation drift. Multiple analyses or statistical measures often converge to similar results, providing corroborating evidence (e.g., Roth & Zohary 2015a,b). On the other hand, slightly different analyses can probe different aspects of the data shedding light on underlying sources of the studied phenomenon (Roth, 2016). When measuring model goodness-of-fit with cvR^2 , we found drift; but when measured with correlation, we did not. These two measures are sensitive to different aspects of the model prediction: while cvR^2 is sensitive to tuning shape, tuning gain, and tuning baseline, Pearson's correlation is insensitive to changes in gain and baseline. Taking this difference into account, we conducted additional analysis to pinpoint the drift as changes in the mean response amplitude. Thus, the two measures yielded complementary results, each providing unique information regarding the stability of neural tuning. We now discuss this point in the Discussion section.

Following the reviewer's suggestion, we reanalyzed the data, limiting the analysis to voxels with $pRF R^2 > 0.75$. For these voxels with better pRF fits, the cross-session cvR^2 values are around 0, i.e., less strongly negative. These voxels also exhibited significant drift. However, we posit that the drift was not limited to visually responsive voxels, and hence selecting voxels according to $pRF R^2$ may be an inappropriate selection criterion.

Figure R8: Analysis limited to voxels with $pRF R^2 > 0.75$. Left: cvR^2 as function of number of intervening sessions between training and test sessions. Colored thin lines, individual subjects; thick

black line, mean across subjects. Right: null distribution of correlation values (gray histogram), and the empirical value (black vertical line).

REVIEWERS' COMMENTS

Reviewer #2 (Remarks to the Author):

In this revision, Roth & Merriam address many of the reviewer criticisms, but unfortunately do not address a principle concern I had with the paper.

My concern is that the main effect they observe over sessions is not a change in tuning, but a change in response amplitude. A change in response amplitude could be entirely explained somewhat trivially by a progressive change in behavioral state (e.g., decreasing arousal, increased fixation breaks). Such a change in behavioral state is entirely within the realm of possibility given that the participants are likely to become progressively less engaged over repeated sessions of passive viewing (see Sadeh & Clopath, 2022). I hate doing this in a second review, but without eye-tracking data to eliminate this confound, I do not feel confident of the conclusions. Many in the field, including myself, would be excited to see these results reproduced in humans, but I am not convinced this is the right data set for doing so.

Minor concerns:

1. Figure R8 appears to be missing the null distribution?
2. Minor point, but I am not sure I agree with the authors that 3 images presented many weeks apart would cause repetition suppression. Repetition suppression typically only occurs with images shown over repeated presentations close in time.

REVIEWERS' COMMENTS

Reviewer #2 (Remarks to the Author):

In this revision, Roth & Merriam address many of the reviewer criticisms, but unfortunately do not address a principle concern I had with the paper.

My concern is that the main effect they observe over sessions is not a change in tuning, but a change in response amplitude. A change in response amplitude could be entirely explained somewhat trivially by a progressive change in behavioral state (e.g., decreasing arousal, increased fixation breaks). Such a change in behavioral state is entirely within the realm of possibility given that the participants are likely to become progressively less engaged over repeated sessions of passive viewing (see Sadeh & Clopath, 2022). I hate doing this in a second review, but without eye-tracking data to eliminate this confound, I do not feel confident of the conclusions. Many in the field, including myself, would be excited to see these results reproduced in humans, but I am not convinced this is the right data set for doing so.

The reviewer is concerned that the changes in response amplitude might be a trivial result of behavioral changes, such as decreasing arousal or increasing eye movement frequency.

As we laid out in the Discussion section of the revision (*'Mechanisms of representational drift'*), citing Sadeh & Clopath 2022, changes in arousal are indeed one of several plausible mechanisms by which stimulus-evoked responses change over time. In that sense, we entirely agree with the reviewer's comment. However, the assumption that arousal modulates stimulus-evoked response *in humans* is far from trivial. One study that tested this question found no impact of locomotion on stimulus responses measured with EEG (Benjamin et al. 2018). A recent study in macaque (Liska et al., bioRxiv 2022) found that when arousal effects were analyzed using an approach analogous to those used in mice, the results were qualitatively different (negative modulation in monkey vs. positive modulation in mice). These results force one to consider the possibility that effects reported in mice do not generalize to humans.

Understanding exactly how changes in arousal could cause representational drift is nontrivial. Studies in mice typically find that arousal increases neural response gain multiplicatively (Niell & Stryker 2010, Mineault et al. 2016, Erisken et al. 2014). In contrast, we report additive baseline changes. We also report that voxels do not change response amplitude uniformly: some voxels have correlated change while others have anti-correlated changes (Fig 3E).

Other behavioral changes are also possible candidates for the underlying mechanism of representational drift. But again, one must consider how such a change would result in drift. Eye movements would shift the retinal image, rendering our model predictions less accurate. If eye movements increase as the experiment progresses, we would expect the model signal-to-noise ratio to decrease over time, which was not the case (Fig 1E).

Speculating about the mechanism of drift by no means renders the phenomenon ‘trivial’. On the contrary, we believe that discovering the mechanism of drift is necessary in order to push to field forward, enabling us to answer complex questions regarding the role of representational drift in sensory cortex.

Minor concerns:

1. Figure R8 appears to be missing the null distribution?

We have added the figure to the supplementary figures.

2. Minor point, but I am not sure I agree with the authors that 3 images presented many weeks apart would cause repetition suppression. Repetition suppression typically only occurs with images shown over repeated presentations close in time.

As the reviewer states, repetition suppression is typically tested using short-term repetition, separated by seconds or minutes. But just as recent longitudinal studies have uncovered representational drift across longer timescales (i.e., days and weeks) we believe similar approaches should be used to test for long-term repetition suppression. Behavioral studies have indeed found evidence of repetition effects that last months (Milleville et al. 2022). It is likely that repetition suppression effects underlie such long-term behavioral priming. Indeed, a recent study found long-term repetition suppression effects in monkey inferior temporal cortex (Koyano et al. 2023). It is therefore possible that repetition effects are present in longitudinal data, particularly those involving many stimulus repetitions such as in previous representational drift studies (Deitch et al. 2021, Marks & Goard 2021). However, we agree with the reviewer that the relatively few repeated presentations in our study (i.e., 3 repetitions, often spaced many weeks apart) likely to result in modest repetition suppression that likely did not have a substantial impact on our measurement of representational drift.